# Posterior Re-calibration for Imbalanced Datasets

**Junjiao Tian**
Georgia Institute of Technology
jtian73@gatech.edu

**Yen-Cheng Liu**
Georgia Institute of Technology
ycliu@gatech.edu

**Nathaniel Glaser**
Georgia Institute of Technology
nglaser@gatech.edu

**Yen-Chang Hsu**
Georgia Institute of Technology
yenchang.hsu@gatech.edu

**Zsolt Kira**
Georgia Institute of Technology
zkira@gatech.edu

## Abstract

Neural Networks can perform poorly when the training label distribution is heavily imbalanced, as well as when the testing data differs from the training distribution. In order to deal with shift in the testing label distribution, which imbalance causes, we motivate the problem from the perspective of an optimal Bayes classifier and derive a post-training *prior rebalancing* technique that can be solved through a KL-divergence based optimization. This method allows a flexible post-training hyper-parameter to be efficiently tuned on a validation set and effectively modify the classifier margin to deal with this imbalance. We further combine this method with existing likelihood shift methods, re-interpreting them from the same Bayesian perspective, and demonstrating that our method can deal with both problems in a unified way. The resulting algorithm can be conveniently used on probabilistic classification problems agnostic to underlying architectures. Our results on six different datasets and five different architectures show state of art accuracy, including on large-scale imbalanced datasets such as iNaturalist for classification and Synthia for semantic segmentation. Please see https://github.com/GT-RIPL/UNO-IC.git for implementation.

## 1 Introduction

Applications of deep learning algorithms in the real world have fueled new interest in research beyond well-constructed datasets where the classes are balanced and the training distribution faithfully reflects the true (testing) distribution.

However, it is unlikely that one can anticipate all possible scenarios and construct well-curated datasets consistently. Therefore it is important to study robust algorithms that can perform well with such imbalanced datasets and unseen situations during testing. The aforementioned problems can be categorized as distributional shift between the training and testing conditions, specifically:*label prior shift* and *non-semantic likelihood shift*.

In this paper, we focus on **label prior shift**, which can arise when i) the training class distribution does not match the true (test) class distribution due to the inherent difficulty in obtaining samples from certain classes, or ii) the distribution of classes does not reflect their relative importance. For example, the natural world is inherently imbalanced such that a collected dataset may exhibit a long-tailed distribution. Similarly, in semantic segmentation, the pixel percentage of pedestrains does not reflect their importance in prediction (e.g. for safety).

In order to deal with such shift in the testing label distribution, which imbalance causes, a number of methods have been developed such as class-balanced loss (CB) [1] and Deferred Reweighting (DRW) [2], as well as methods that scale the margin in a fixed way as a function of the amount of data per class through a label-distribution-aware loss (LDAM) [2] . However, such methods do not allow for a flexibly trade-off between precision and recall across the classes, and require re-training to target a testing label distribution.

In this paper, we instead motivate the problem from the perspective of an optimal Bayes classifier and derive a *rebalanced posterior* by introducing test priors and showing that it is the optimal *Bayes classifier* on the test distribution under ideal conditions. To compensate for imperfect learning in practice, an approximation can be solved through a KL-divergence based optimization by treating the rebalanced and original posteriors as approximations to a true posterior.

This method, which does not require re-training the original imbalanced classifiers, allows a flexible hyper-parameter to be efficiently tuned on a validation set and effectively modify the classifier margin to target a desired test label distribution. We further combine this method with existing methods dealing with **non-semantic likelihood shift**, which occurs when the representation of samples from a certain class changes due to changes in lighting, weather, sensor noise, modality, etc. We re-interpret these methods from the same Bayesian perspective, and demonstrate that our method can deal with both problems in a unified way.

We demonstrate our method on six datasets and five different neural network architectures, across two different imbalanced tasks: classification and semantic segmentation. Using a toy dataset that allows for visualizations of the classifier margins, we show that our method effectively shifts the decision boundary away from the minority class towards the over-represented class. We then demonstrate that our method can achieve state of the art results on imbalanced variants of CIFAR-10 and CIFAR-100, and can scale to larger datasets such as iNaturalist which has extreme imbalance. For semantic segmentation, which is an inherently imbalanced task, we show that our unified method can support both imbalance and likelihood shift in the form of unknown weather conditions encountered during testing (but not during training).

In summary, the main contributions of the paper are the following:

- We derive a principled *imbalance calibration* algorithm for models trained on imbalanced datasets. The method requires no re-training to target new test label distributions and can flexibly trade-off between precision and recall on under-represented classes via a single hyper-parameter.

- We introduce an efficient search algorithm for this hyper-parameter. Unlike cost sensitive learning, an optimal hyperparameter can be searched efficiently on a validation set post-training.

- We test the algorithm on six datasets and five different architectures and outperforms state-of-the-art models on classification accuracy (recall) across all tasks and models while maintaining good precision.

- We further combine our method with non-semantic likelihood shift methods, re-motivate it from the Bayesian perspective, and show that we can tackle both problems in a unified way on a RGB-D semantic segmentation dataset with unseen weather conditions. We show significant improvement on mean accuracy while maintaining good mean IOU performance with both qualitative and quantitative results.

## 2 Related Work

In this paper, we focus on prior (label) distribution shift resulting from various degrees of imbalance in the training label distributions, and testing on a different distribution or emphasizing a different distribution in the testing metrics (e.g. class-averaged accuracy). We therefore summarize methods for dealing with such imbalance.

**Imbalance - Data Level Methods** The simplest methods for dealing with label imbalance during training are random under-sampling (RUS) which discards samples from the majority classes and random over-sampling (ROS) which re-samples from the minority classes [3]. While ROS is infeasible when the data imbalance is extreme, RUS tends to overfit the minority classes [4]. Synthetic

generation [5] or interpolation [6] to increase the number of samples in the minority class are also used. However, these methods are sensitive to imperfections in the generated data.

**Imbalance - Algorithm-Level Methods** The majority of work in this category modify the training procedure by introducing cost-sensitive losses. For example, *median frequency balancing* has been used in dense prediction tasks such as surface normal and depth prediction [7] as well as semantic segmentation [8]. DRW [2] is a variant of the frequency balancing algorithm. It re-weights the loss function at a latter training stage not from the beginning. We specifically compare our method to three state-of-the-art cost sensitive losses LDAM [2], Class-Balanced Loss (CB) [1], and Focal Loss (FL) [9]. LDAM derives a generalization error bound for the imbalanced training and proposes a margin-aware multi-class weighted cross entropy loss. CB motivates the concept of *effective* number of samples and derives a weighted cross entropy loss with approximation. FL is another weighted cross entropy loss with a sample-specific weight $(1 - p_i)^\gamma$, where $p_i$ is the model output probability for class $i$. A contemporary work Bilateral-Branch Network (BNN) [10] proposes a two branch approach with one branch learning the original distribution and the the other learning a rebalanced distribution. Our method can be combined with algorithm-level methods to yield better performance.

**Imbalance - Classifier Level Methods** The most common method in this category is *thresholding* or *post scaling*. The process happens at the test phase and changes the output class probabilities. The simplest variant compensates for prior class probabilities, i.e. dividing the output for each class by its training set prior probabilities. This method can often improve the performance for imbalanced classification [11] [12]. The same technique under the name *maximum likelihood* [13] is also used in semantic segmentation which is a naturally class-imbalanced task. While this decision rule was able to improve recall on minority classes, it deteriorates overall performance by introducing substantially more false detection. Our method belongs to this category and improves on previous works. For a more comprehensive review on dataset imbalance in deep learning, we refer readers to [4].

## 3 Method

### 3.1 Background: A Unified Perspective on Label Prior and Non-Semantic Likelihood shift

Let $P_s(X, Y)$ define a training (source) distribution and $P_t(X, Y)$ define a test (target) distribution where $X$ is the input and $Y \in \mathbb{C} = \{1, ..K\}$ is the corresponding label. In a multi-class classification task, the final decision on the target distribution is often made by following the Bayes decision rule:

$$y^* = \arg\max_{y \in \mathbb{C}} P_t(y|x) = \arg\max_{y \in \mathbb{C}} \frac{f_t(x|y)P_t(y)}{P_t(x)} = \arg\max_{y \in \mathbb{C}} f_t(x|y)P_t(y) \qquad (1)$$

where $f_t(x|y)$ is the class conditional probability density function and $P_t(y)$ is the prior distribution.

*Label prior shift* refers to the prior (label) distribution changing between training and testing, i.e, $P_s(Y) \neq P_t(Y)$; in this paper, we focus on the case where the training distributions have varying degrees of imbalance (e.g. long-tailed) and the testing distribution shifts (e.g. is uniformly distributed). Note, however, that the method we develop is not limited to this case. Note also that some metrics during testing implicitly emphasize a uniform distribution over labels, regardless of the actual label distribution in the testing dataset; see Appendix Section 6.3 for a proof for one popular class of metrics, namely class-averaged metrics.

*Non-semantic likelihood shift* refers to shift without introducing new semantic labels such as sensor degradations, changes in lighting, or presentation of the same categories in a different modality, i.e. $f_s(X|Y) \neq f_t(X|Y)$.

In this paper, we focus on the label prior shift, commonly known as class imbalance. In sec. 3.2, to deal with prior label shift, we motivate the problem from the perspective of the optimal Bayes classifier and derive a *prior rebalancing* equation and a KL-divergence based optimization method. In addition, we re-motivate temperature scaling [14] for dealing with likelihood shift, starting with a similar Bayesian perspective, as a *likelihood flattening* process instead of as a confidence calibration technique. In the same section, we propose a unified algorithm to simultaneously handle both types of shifts.

## 3.2 Imbalance Calibration (IC) with Prior Rebalancing

### 3.2.1 Theoretical Motivation

To motivate the framework for dealing with learning in an imbalanced setting, we start from the assumption that a discriminative model can learn the posterior distribution $P_s(Y|X)$ of the source dataset perfectly. The corresponding *Bayes Classifier* is defined as the following.

$$h_s(x) = \arg\max_{y \in \mathbb{C}} P_s(Y = y|X = x) \tag{2}$$

The classifier $h_s(x)$ is the optimal classifier on the training distribution, $P_s(X, Y)$. Now let's assume that the test distribution has the same class conditional likelihood as the training set, i.e. $f_t(X|Y) = f_s(X|Y)$ and differs only on the class priors $P_t(Y) \neq P_s(Y)$. We show the following result:

**Theorem 1** *Given that $h_s(x)$ is the Bayes classifier on $P_s(X, Y)$,*

$$h_t(x) = \arg\max_{y \in \mathbb{C}} \frac{P_s(y|x)P_t(y)}{P_s(y)}, \tag{3}$$

*is the optimal Bayes classifier on $P_t(X, Y)$ where $f_t(X|Y) = f_s(X|Y)$ and $P_t(Y) \neq P_s(Y)$ and the Bayes risk is $R(h_t) = P(h_t(x) \neq y)$.*

This partition defines the decision region of a classifier for $K$ classes.

$$1 - R(h_t) = P(h_t(x) = y) = \sum_{k=1}^{K} P_t(y = k)P(h_t(x) = k|y = k) \tag{4}$$

$$= \sum_{k=1}^{K} P_t(k) \int_{\Gamma_k(h_t)} f_t(x|k)dx = \sum_{k=1}^{K} P_t(k) \int_{\Gamma_k(h_t)} f_s(x|k)dx$$

$$= \sum_{k=1}^{K} P_t(k) \int_{\Gamma_k(h_t)} \frac{P_s(k|x)f_s(x)}{P_s(k)}dx = \int_{\mathbb{R}^D} \left( \sum_{k=1}^{K} \mathbb{I}_{\Gamma_k(h_t)}(x) \frac{P_s(k|x)P_t(k)}{P_s(k)} f_s(x) \right) dx$$

where $\mathbb{I}_{\Gamma_k(h)}(x) = 1, \forall x \in \Gamma_k(h)(x)$ and 0 otherwise. To avoid notational clutter, we abbreviate $y = k$ as $k$. By choice of the decision rule $h_t(x)$ defined in Thm. 1, the function inside the integral is at its maximum and any other decision rules will result in higher risk. We note that the rebalancing technique in Thm. 1 is not new and has been studied before in [15]. Our contribution is proving its optimality under the *ideal setting* with Bayes risk.

The theorem sheds light on the weakness of this approach: in reality, we do not have access to either $P_t(X, Y)$ or $P_s(X, Y)$. We merely have samples from those hidden distributions. In other words, we do not have perfect modeling nor learning of the generating distribution. We learn an approximation of the distribution according to *relative entropy* between a parametrized distribution $P_d(Y|X)$, denoting the learned *discriminative* posterior, and $P_s(Y|X)$. Therefore, the resulting classifier with proper substitution to Thm. 1,

$$\tilde{h}_t(x) = \arg\max_{y \in \mathbb{C}} \frac{P_d(y|x)P_t(y)}{P_s(y)}, \tag{5}$$

is not optimal and the corresponding partition $\Gamma(\tilde{h}_s)$ is not the optimal decision boundary on the test distribution. We will now show that we can obtain a better classifier than $\tilde{h}_t(x)$ by exploring some observations.

We introduce a terminology *rebalanced posterior*, $P_r(y|x)$ to describe eq. 5 because the discriminative posterior $P_d(y|x)$ is rebalanced by $P_t(y)/P_s(y)$. As we have discussed, the parametrization and learning of the training distribution are not perfect. It is unlikely that a direct application of theorem 1 leads to a good classifier on the test distribution. The weighting $P_t(y)/P_s(y)$ in eq. 5 might be too large or too small for different setups. For example, in semantic segmentation, directly using eq. 5 results in excessive false positives for small classes as reported by [13]. We also show in the experiment section that eq. 5 yields inferior performance on imbalanced classification datasets.

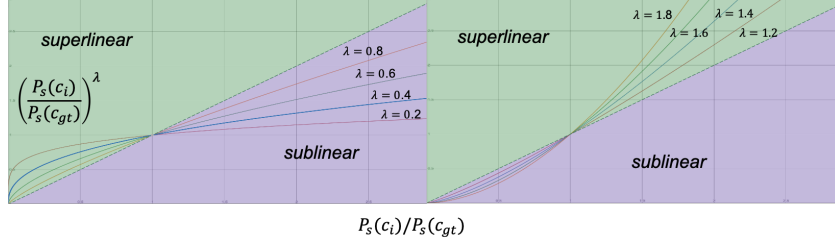

Figure 1: **Left:** log-linear dependency on *source prior ratio* with $\lambda \leq 1$. **Right:** log-linear dependency on *source prior ratio* with $\lambda > 1$. $x$ axis is the ratio $P_s(c_i)/P_s(c_{gt})$ and $y$ axis denotes the ratio raised to the power of $\lambda$. The amplification effect of the source prior ratio is reduced when $\lambda \leq 1$ and increased when $\lambda > 1$

**Hypothesis 1** *A better classifier can be found by finding a trade off between the discriminative classifier $P_d(y|x)$ and the derived rebalanced classifier $P_r(y|x)$.*

Intuitively, if the weighting $P_t(y)/P_s(y)$ is too large and $P_r(y|x)$ biases too much towards small classes, we might want to weigh in the original prediction $P_d(y|x)$ where no weighting is applied; if the weighting is too small, we can encourage a stronger weighting to move further away from $P_d(y|x)$. Based on this intuition, a natural distance metric for distributions is KL-divergence (or relative entropy). Therefore, we propose the following optimization problem based on minimizing the KL-divergence between the final *calibrated posterior* $P_f(y|x)$ and $P_d(y|x), P_r(y|x)$.

$$P_f^*(y|x) = \arg\min_{P_f}(1 - \lambda)\mathcal{KL}(P_f, P_d(y|x)) + \lambda\mathcal{KL}(P_f, P_r(y|x)) \tag{6}$$

where $\lambda \geq 0$ is a hyperparameter. The optimization problem tries to find an interpolated distribution between $P_d(y|x)$ and $P_r(y|x)$ with a regularizing parameter $\lambda$. The information-theoretic justification for this formulation is that assuming $P_f$ represents the true distribution which we want to approximate, $\mathcal{KL}(P_f, P_d(y|x)) + \mathcal{KL}(P_f, P_r(y|x))$ captures the number of bits lost when encoding $P_f$ with both $P_d$ and $P_r$. It is clear that when $\lambda = 0$ we recover the discriminative classifier and when $\lambda = 1$ we have the rebalanced classifier. More generally, there exists a closed form solution to this optimization problem [16].

$$P_f^*(y|x) = \frac{1}{Z(x)}\left(P_d(y|x)^{1-\lambda}P_r(y|x)^{\lambda}\right), \tag{7}$$

where $Z(x)$ is the normalization factor. This completes an extremely simple calibration method with only a single hyperparameter for class imbalance in classification. It can effectively adjust a classifier's decision boundary and outperform more sophisticated training methods on image classification tasks. Further more, unlike many existing class imbalance methods, we will show that the proposed method can adjust the trade-off between precision and recall. This is especially important for semantic segmentation which uses Intersection over Union (IOU) as criterion.

### 3.2.2 Analysis of the Rebalancing Equation

We use the *odd ratio* form for analysis. Odd ratio has been used in logistic regression [17], where it is defined as the ratio of probability of success and probability of failure. Here odd ratio denotes ratio of the *calibrated posteriors*. Let $c_{gt}$ denote the ground truth class, $c_i$ denote one of the other classes. We rewrite Eq. 7 as the following ( detailed derivation is in appendix 6.1):

$$\frac{P_f^*(c_{gt}|x)}{P_f^*(c_i|x)} = \frac{P_d(c_{gt}|x)}{P_d(c_i|x)}\left(\frac{P_s(c_i)}{P_s(c_{gt})}\right)^{\lambda}\left(\frac{P_t(c_{gt})}{P_t(c_i)}\right)^{\lambda} \tag{8}$$

For a ground truth class to be the selected (argmax) output, the odd ratio in eq. 8 needs to be greater than $1 \ \forall i \neq gt$. We first analyze the dependency on $\lambda$ by setting the test priors, $P_t(y)$, equal for all classes.

It is clear from Eq. 8 that if $c_{gt}$ is a minority class then the ratio of priors under $\lambda$, $P_s(c_i)/P_s(c_{gt})$, denoted *source prior ratio*, is greater than 1 and when $c_{gt}$ is a large class it is smaller than 1. Intuitively, this prior ratio amplifies the odd ratio for minority classes and downweights the odd ratio

for large classes. However, as pointed out in [13], this scaling can be far too aggressive when $\lambda = 1$. In other words, the prior ratio becomes too large when $c_{gt}$ is a minority class and too small when $c_{gt}$ is a large class.

As shown in the left part of Fig. 1, with $\lambda \leq 1$ when the *source prior ratio* is greater than 1 in the case of $c_{gt}$ being a minority class, the growth is *sublinear* and when the *source prior ratio* is smaller than 1, the growth is *superlinear*. In the right section of fig. 1, we also see that the dependency is different when $\lambda \geq 1$. When $c_{gt}$ is a minority class, the growth is *superlinear* and when $c_{gt}$ is a majority class, the growth is *sublinear*. **In other words, the amplification effect of the source prior ratio is reduced when $\lambda \leq 1$ and increased when $\lambda \geq 1$.** As we will show empirically, by finding a good $\lambda$ on a validation set, we can strike a balance between recall and precision between large and small classes. The analysis validates our hypothesis that a better classifier can indeed be found through the trade-off of $P_d(y|x)$ and $P_r(y|x)$.

### 3.2.3 Search Algorithm and Time Complexity for $\lambda$

An efficient algorithm exists for searching $\lambda$ based on an empirical observation of the dependency between accuracy and $\lambda$. Empirically, we found that performance metrics evaluated on a validation set vary with $\lambda$ in a concave manner with a unique maximum. A modified *binary search* algorithm can find a maximum with a specified precision with $\mathcal{O}(\log N)$ time complexity where $N$ is the number of $\lambda$s in the search range. Due to space constraint, please refer to the appendix for the complete algorithm 6.2. Please refer to the experiments in sec. 4.1 for more detail.

### 3.3 Confidence Calibration with Likelihood Flattening

In the previous section, we explored a probabilistic way to deal with prior shift. In this section, we similarly look at likelihood shift and cast a different perspective on an existing technique, temperature scaling [14], as a *likelihood flattening* process which *implicitly* increases the uncertainty in the likelihood $P_d(X|Y)$. The reasoning is hypothetical because a discriminative model does not learn a likelihood explicitly. Temperature scaling has been used in uncertainty calibration [18]. The basic temperature scaling takes the following form,

$$\mathbf{P}(Y|x) = \text{Softmax}\left([l_1, ..., l_{N_c}] * \delta\right), \tag{9}$$

where $[l_1, ..., l_{N_c}]$ are the pre-softmax logits and $\delta$ is a scalar.

Graphically, smaller $\delta$ will result in a flatter softmax output and larger $\delta$ will result in a more peaked softmax distribution. In Eq. 1, we showed that $y^* = \arg\max_{y \in \mathbb{C}} P_t(y|x) = \arg\max_{y \in \mathbb{C}} f_t(x|y)P_t(y)$. Considering the equal prior situation, then we can remove $P_t(y)$ from the previous equation. Now, flattening the softmax probability, $P_t(y|x)$, is equivalent to flattening the likelihood, $f_t(x|y)$. We formalize the discussion in the following hypothesis.

**Hypothesis 2** *In light of the decomposition in Eq. 1, temperature scaling in a softmax-activated discriminative model is numerically equivalent to flattening the class conditional likelihood $f(x|y)$ $\forall y \in \mathbb{C}$ implicitly and making the likelihood of belonging to any class more likely.*

This behavior of temperature scaling is desirable especially in a multi-modal fusion setting. For example, if one modality is compromised, temperature scaling can flatten the output distribution from that modality to raise its uncertainty in the sense of entropy. More importantly, a flattened distribution has little power to affect the decision from other modalities.

UNO [19] introduced an uncertainty-aware temperature scaling factor $\delta(x)$ which depends on a network's input $x$. The uncertainty-aware scalar correlates with uncertainty in an input image. Intuitively, if an input image is degraded, $\delta(x)$ as the temperature parameter will be a scalar less than 1 and thus "softens" the distribution and makes the model less confident in its prediction.

Following the hypothesis, we view the uncertainty-aware temperature scaling in UNO as a likelihood flattening method. Together with the prior rebalancing method from the previous section, we develop a unified *multi-modal fusion* algorithm Alg. 1 for prior and likelihood shift and apply it to multi-modal tasks. We use *noisy-or* operation as the final fusion layer as in UNO [19]. We demonstrate the performance of the algorithm on a RGB-D semantic segmentation task in Seq. 4.4.2. Please also see Appendix 6.6 for qualitative results.

**Algorithm 1:** UNO-IC: multi-modal fusion algorithm for prior and likelihood shift

---

**Data:** $\{x\} \in D_{test}$

**Result:** $\arg\max_y \mathbf{P}_f(Y|x)$

1 **Initialize:** $\mathbf{L}_d^m(Y|X) \quad \forall m \in \{1, ..., M\}$      ▷ $\mathbf{L}_d^m$ is the pre-softmax logits of *discriminative* model $m$

2 **for** $m = 1, .., M$ **do**

3      $\mathbf{P}_d^m(Y|x) = \text{Softmax}\left(\mathbf{L}_d^m(Y|x) * \delta_m(x)\right)$      ▷ *Likelihood Flattening* 3.3 using $\delta_m(x)$ in [19]

4      $\mathbf{P}_r^m(Y|x) = \frac{\mathbf{P}_d^m(Y|x)\mathbf{P}_r^m(Y)}{\mathbf{P}_s^m(Y)}$      ▷ *Prior Rebalancing* 3.2 using Eq. 5

5      $\mathbf{P}_f^m(Y|x) = \frac{1}{Z(x)}\left(\mathbf{P}_d^m(Y|x)^{1-\lambda}\mathbf{P}_r^m(Y|x)^\lambda\right)$      ▷ *Calibrated Posterior* using Eq. 7

6 $\mathbf{P}_f(Y|x) = \frac{1}{Z(x)}\left(1 - \prod_m 1 - \mathbf{P}_f^m(Y|x)\right)$      ▷ *Noisy-Or* fusion

---

Table 1: **Summary of datasets and architectures:** Imbalance Ratio is the ratio of class size between the largest and smallest class.

| | Dimension | Num of Classes | Max. Class Size | Imbalance Type | Imbalance Ratio | Arch. |
|---|---|---|---|---|---|---|
| Two Moon | $\mathcal{R}^2$ | 2 | $\sim 2,500$ | Step | 9 | 3-layer FCN |
| Circle | $\mathcal{R}^2$ | 2 | $\sim 2,500$ | Step | 9 | 3-layer FCN |
| Cifar10 | $\mathcal{R}^{32\times32}$ | 10 | $5,000$ | Step/Exp | 100/10 | Resnet32 [20] |
| Cifar100 | $\mathcal{R}^{32\times32}$ | 100 | $5,000$ | Step/Exp | 100/10 | Resnet32 [20] |
| iNaturalist18 [21] | $\mathcal{R}^{299\times299}$ | $8,142$ | $1,000$ | Long-tail | 500 | IncepV3 [22]/Resnet50 [20] |
| Synthia [23] | $\mathcal{R}^{768\times384}$ | 14 | $1.9\times10^9$ | Long-tail | $1.3\times10^4$ | DeepLab [24] |

## 4 Experiments

We conduct experiments to test our imbalance calibration (IC) algorithm described in sec. 3.2 on six datasets with five different architectures. A summary of datasets and architectures used for each one is shown in table 1.

### 4.1 Toy Dataset

To visualize the effect of our imbalance calibration algorithm on decision boundaries, we generate two binary imbalanced datasets for classification, *two moon* and *circle* and train a 3-layer fully connected classifier with logistic loss on each. Fig. 2 shows an overlay of the input data and decision boundary with different $\lambda$s and the accuracy vs. lambda plots for both datasets. The original decision boundary $\lambda = 0.0$ cuts through the minority class, and varying $\lambda$ shifts the decision boundary away from the minority class towards the larger class. We can find a better classifier empirically by searching $\lambda$ values on a validation set. The accuracy vs. lambda curves exhibit concavity as shown on the right side of Fig. 2. This observation motivates an efficient *modified binary search algorithm* for $\lambda$ in Appendix Sec. 6.2. The same concavity is also observed for more complex tasks subsequently.

### 4.2 Imbalanced CIFAR Classification

We compare our methods with the state-of-the-art methods [1] [2] [9] [10] on CIFAR-10 and CIFAR-100 dataset with two different imbalance types: long-tailed imbalance [1] and step imbalance [12] with two different imbalance ratios for each imbalance type. Following [2], we present the validation error in Table 2. We observe that Focal [9] is not as effective against imbalance and this is also

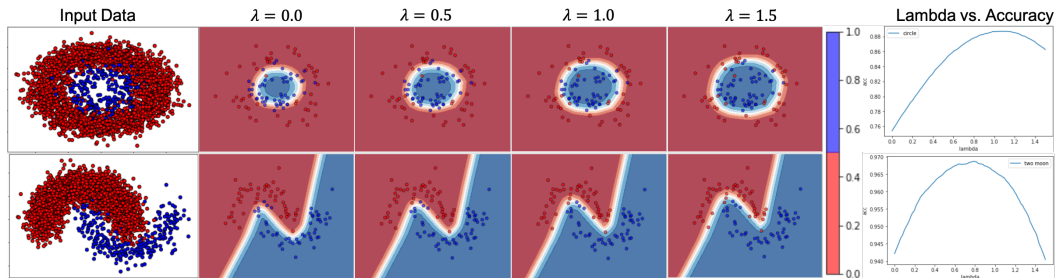

Figure 2: **Top:** Circle **Bottom:** Two moon. The red class is the marjority class. S $\lambda$ increases, it effectively shifts the decision boundary away from the minority class towards the large class.

Table 2: **Top1 validation error**↓ * indicates the reported results from [10]. Our model (-IC) achieves the best performance overall. CE is a baseline trained with unmodified cross-entropy loss. CB refers to class balanced loss in [1]. Combining our method with techniques such as Deferred Reweighting (DRW) [2] yields the best performance.

| Dataset | Imbalanced CIFAR-10 | | | | Imbalanced CIFAR-100 | | | | |
|---|---|---|---|---|---|---|---|---|---|
| Imbalance Type | long-tailed | | step | | long-tailed | | step | | AVE |
| Imbalance Ratio | 100 | 10 | 100 | 10 | 100 | 10 | 100 | 10 | |
| CE | 28.48 | 13.47 | 34.01 | 14.54 | 62.16 | 43.21 | 61.28 | 45.33 | 37.70 |
| Focal[9] | 26.62 | 13.52 | 34.89 | 14.92 | 62.02 | 43.60 | 61.49 | 45.62 | 38.17 |
| LDAM[2] | 26.39 | 13.77 | 34.42 | 15.33 | 59.70 | 43.52 | 60.59 | 49.20 | 37.91 |
| CB CE[1] | 26.77 | 12.37 | 37.43 | 14.44 | 67.93 | 44.15 | 80.40 | 47.28 | 41.35 |
| CB Focal[1] | 25.93 | 13.06 | 36.07 | 14.8 | 68.08 | 45.17 | 78.14 | 46.84 | 41.40 |
| CE-DRW[2] | 23.78 | 11.85 | 29.05 | 12.79 | 58.64 | 41.53 | 55.70 | 40.65 | 34.25 |
| Focal-DRW[2] | 25.73 | 12.00 | 27.37 | 11.82 | 60.47 | 43.35 | 61.63 | 41.93 | 33.9 |
| LDAM-DRW[2] | 23.38 | 12.12 | 22.69 | 12.68 | 57.77 | 42.52 | 54.30 | 43.49 | 33.62 |
| BNN[10] | 20.18* | 11.58* | – | – | 57.44* | 40.88 | – | – | – |
| CE-IC (Ours) | 20.14 | 11.37 | 25.61 | 11.34 | 59.08 | 41.91 | 53.48 | 40.64 | 32.95 |
| Focal-IC (Ours) | 19.84 | 11.99 | 25.68 | **10.99** | 58.35 | 42.01 | **52.73** | 40.70 | 32.79 |
| Focal-DRW-IC (Ours) | 21.63 | 11.80 | 21.63 | 11.48 | 59.43 | 43.35 | 56.89 | 41.93 | 33.52 |
| CE-DRW-IC (Ours) | **18.91** | **11.44** | **21.45** | 11.42 | **56.89** | 41.44 | 54.40 | **40.07** | **32.00** |

Table 3: **Validation error↓ on iNaturalist2018.** Our method (-IC) yields the best performance and is effective for extreme large number of classes and severe imbalance. We use 3-fold cross validation for tuning $\lambda$ and testing. Subscript value represents the $\lambda$ for that experiment. * indicates the reported results from [10]

| | CE | CE-DRW[2] | Focal[9] | Focal-DRW[2] | LDAM[2] | LDAM-DRW[2] | BNN[10] | CE-IC | CE-DRW-IC |
|---|---|---|---|---|---|---|---|---|---|
| InceptionV3 [22] | 40.08 | 35.35 | 39.68 | 35.79 | 41.81 | 37.03 | – | **$34.16 \pm 0.03$** | **$34.22 \pm 0.11$** |
| Resnet50 [20] | 38.00 | 33.73 | 38.33 | 34.38 | 39.62 | 35.42* | 33.71* | **$32.16 \pm 0.41$** | **$32.06 \pm 0.38$** |

observed in the same set of experiments in [2]. BNN [10] achieves comparable performance with more complicated training strategy and architecture. Even though LDAM [2] improves the classification performance, DRW gives the most improvement. As a post-calibration method, our method can be combined with different learning losses and learning schedules to further improve performance. Our method combined with DRW and standard cross entropy loss, CE-DRW-IC, yields the best performance. The hyperparameter $\lambda$ is searched on the validation set. Please refer to the Appendix Sec. 6.4 for the exact values.

## 4.3 iNaturalist Dataset Classification

iNaturalist2018 [21] is a large classification dataset for species with 437,513 training images and 24,426 validation images. We randomly split the validation set into three subsets and use 3-fold cross validation for tuning and testing our IC method. We compare our method with Focal [9], LDAM [2], DRW [2] and BNN [10] in Table 3. Again, IC methods yield the best performance. It outperforms BNN [10] which has a much more complicated training strategy. The result demonstrates the effectiveness of our method for extremely large number of classes and severe imbalance.

Table 4: **Test mean IOU, mean accuracy on Synthia in distribution splits** (RGB only). Our method (IC) with cross entropy or Focal loss achieves significant improvement in mean accuracy while maintaining competitive mean IOU.

| | CE (Unweighted) | CE (Median Freq.) | CE(Max. Freq.) | LDAM[2] | Focal[9] | CE-IC (Ours) | Focal-IC (Ours) |
|---|---|---|---|---|---|---|---|
| mIOU ↑ | 84.48 | 76.85 | 76.84 | 71.28 | 82.10 | **$84.71_{0.2}$** | $82.13_{0.1}$ |
| mACC ↑ | 88.59 | 97.89 | **97.91** | 96.87 | 87.49 | $94.93_{0.2}$ | $92.03_{0.1}$ |

Table 5: **Test mean IOU and mean accuracy (mIOU | mACC) of the joint algorithm [1] on Synthia out-of-distribution splits**.The listed conditions have not been encountered during training.

| | mIOU\|mACC | Fog | Fall | Winter | WinterNight | Rain | RainNight | SoftRain | AVE. |
|---|---|---|---|---|---|---|---|---|---|
| Baseline | SoftAve | 81.96\|85.42 | 81.78\|85.33 | 78.37\|82.61 | 79.46\|83.31 | 70.68\|73.94 | 78.96\|82.37 | 78.02\|81.22 | 78.46\|82.03 |
| | UNO[19] | 82.05\|85.49 | 81.85\|85.41 | 78.37\|82.68 | 79.37\|83.30 | 74.28\|77.61 | 79.51\|83.04 | 78.41\|81.68 | **79.12**\|82.74 |
| IC | $\lambda = 0.4$ | 81.33\|92.73 | 80.87\|93.68 | 78.07\|90.49 | 79.01\|91.57 | 72.92\|79.04 | 78.76\|89.16 | 78.86\|88.80 | 78.55\|87.92 |
| UNO-IC | $\lambda = 0.4$ | 81.14\|93.04 | 80.65\|93.91 | 77.77\|90.88 | 78.69\|92.10 | 74.74\|83.23 | 78.53\|90.05 | 78.33\|89.91 | 78.55\|**90.45** |

## 4.4 Synthia Multimodal Semantic Segmentation

For RGB-D semantic segmentation, we use the Synthia-Sequences dataset [25] which is a photorealistic synthetic dataset of traffic scenarios under different weather, illumination, and season conditions. The hyperparameter $\lambda$ is tuned on a validation split and the algorithm is tested on a test set.

### 4.4.1 RGB Imbalance Experiment

Unlike previous classification problems which use only accuracy as the performance metric, semantic segmentation uses both mean intersection over union (IOU) and mean accuracy. Mean IOU considers the effect of false positive predictions while mean accuracy only considers true positives. The difference between the two metrics can be seen as the recall-precision trade-off. Table 4 shows the full evaluation against frequency weighted loss, Focal loss [9] and LDAM [2] on the in-distribution test splits. While the median and max frequency weighted losses achieve the best mean accuracy, their mean IOU performance drops considerably. LDAM [2] resembles the performance of the frequency losses with high accuracy but low IOU because it also emphasises the minority classes by maintaining a fixed margin. Focal loss [9], on the other hand, resembles the performance of unweighted cross-entropy loss because it adopts a per-example weighting scheme and does not explicitly reweight a certain class. BNN [10] does not generalize to segmentation due to its undersampling strategy for training a rebalancing branch. Overall, our method CE-IC and Focal-IC which use imbalance calibration on top of models trained with cross entropy and focal loss achieve significantly improved mean accuracy while maintaining a good mean IOU. Further analysis of the performance characteristics is in Appendix Sec. 6.5.

### 4.4.2 RGBD Fusion Experiment

To demonstrate the proposed unified Alg. 1 for prior and likelihood shift, we conduct RGB-D multimodal semantic segmentation experiments on Synthia out-of-distribution splits which consist of seven weather conditions not seen during training. In this experiment, we compare to a simple baseline, SoftAve, which simply adds RGB and Depth predictions and averages the two, and UNO [19] which is designed for unseen degradations. In Table 5, we report both mean IOU and mean accuracy. While UNO achieves the best mean IOU, our method combined with UNO results in significant improvement in mean accuracy with negligible drop in mean IOU. This experiment demonstrates the effectiveness of our algorithm against extreme dataset imbalance and unseen degradations. We also show qualitative results in Appendix Sec. 6.6. Our method UNO-IC is able to identify small objects in the distance such as poles and pedestrians even under severe visual degradations.

## 5 Conclusion

We proposed a unified perspective and algorithm to deal with label prior shift, and combine it with methods for non-semantic likelihood to tackle both types of shifts simultaneously. The major contribution is a novel imbalance calibration technique which is motivated from the concept of optimal Bayes classifier and optimization of KL divergence. To accompany the formula, we introduced a modified binary search algorithm for the hyperparameter (lambda) based on the empirical observation of a concave performance function as it is varied. Our method is also very computationally efficient because the hyperparameter is tuned during the validation stage whereas most SOTA methods require retraining. The final algorithm can be used in probabilistic classification tasks regardless of underlying architectures. We have shown the generality of the imbalanced calibration technique on six datasets and a number of different architectures, showing effectiveness of the unified algorithm against both extreme dataset imbalance and unseen degradations in multi-modal semantic segmentation.

## Acknowledgement

This work was supported by ONR grant N00014-18-1-2829.

**Broader Impact**

Our work discusses a fundamental problem in deep learning: data imbalance. Data imbalance can be safety-critical in many applications such as self-driving. Our approach allows an algorithm to give more attention to pedestrians and small obstacles. Additionally, our unified algorithm aims to provide stable performance under severe visual degradation which is also critical for safty considerations.

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
