[Supplementary Material]

# 6 Appendix

## 6.1 Odd Ratio Derivation

In Sec. 3.2.2, we used *odd ratio* to analyze the behavior of the proposed imbalance calibration method. Here we provide the full derivation of the odd ratio Eq. 8 from Eq. 7 and Eq. 7.

$$
\begin{aligned}
\frac{P_f(c_{gt}|x)}{P_f(c_i|x)} &= \frac{1/Z \left(P_d(c_{gt}|x)^{1-\lambda} P_r(c_{gt}|x)^{\lambda}\right)}{1/Z \left(P_d(c_i|x)^{1-\lambda} P_r(c_i|x)^{\lambda}\right)} \\
&= \frac{P_d(c_{gt}|x)^{1-\lambda} \left(P_d(c_{gt}|x)P_t(c_{gt})/P_s(c_{gt})\right)^{\lambda}}{P_d(c_i|x)^{1-\lambda} \left(P_d(c_i|x)P_t(c_i)/P_s(c_i)\right)^{\lambda}} \\
&= \frac{P_d(c_{gt}|x)}{P_d(c_i|x)} \left(\frac{P_s(c_i)}{P_s(c_{gt})}\right)^{\lambda} \left(\frac{P_t(c_{gt})}{P_t(c_i)}\right)^{\lambda}
\end{aligned}
\tag{10}
$$

where $c_{gt}$ denotes the ground truth class, $c_i$ denotes one of the other classes. $P_f$, $P_d$ and $P_r$ denote the final calibrated posterior, the original discriminative posterior and the rebalanced posterior respectively.

## 6.2 Search Algorithm for $\lambda$

Efficient algorithm exists for searching $\lambda$ based on empirical observation of dependency on $\lambda$. Empirically, we found that performance metrics evaluated on a validation set vary with $\lambda$ in concave manner with a unique maximum. In other words, a function of $\lambda$ first increases and then decreases. A modified *binary search* algorithm can find a maximum with a specified precision with $O \log(n)$ time complexity where $n$ is the number of $\lambda$s in the search range. We outline the algorithm in a Pythonic pseudo code format implemented with recursion. To assist efficient search, we define three hyperparameters in this algorithm, *Low*,*High* and *Prec* which is the lower bound, upper bound and quantization precision of $\lambda$ to be searched. Based on our observation,the parameters are set to 0.0, 2.0 and 0.1 respectively. Another notation introduced is *metric*() which denotes a metric function that takes $\lambda$ as input and evaluates a model with the given $\lambda$ on a validation set. *metric*() returns a scalar representing the performance of the model with the current $\lambda$. The full algorithm is presented in Alg. 2.

## 6.3 Imbalance Due to Evaluation Metrics

We define two types of label prior shift: explicit and implicit. The shift occurs as a result of imbalance in the training data and they will lead to poor performance measured on certain metrics. We will discuss them in the general case of multi-class classification.

**Explicit label prior shift** refers to the prior (label) distribution changing between training and testing, i.e, $P_s(Y) \neq P_t(Y)$ following the definition of prior shift in [26]; in this paper, we focus on the case where the training distributions have varying degrees of imbalance (e.g. long-tailed) and the testing distribution shifts (e.g. is uniformly distributed, $Y \sim \mathbb{U}(1/K)$). To see that the imbalance causes label prior shift which in turn, causes performance drop, let's analyze the popular metric, accuracy used in image classification.

$$
ACC(X,Y) = \frac{1}{N} \sum_{k=1}^{K} \sum_{i=1}^{N} \mathbb{I}(h(x_i)=k, y_i=k) = \sum_{k=1}^{K} \frac{N_k}{N} \left(\frac{1}{N_k} \sum_{i=1}^{N} \mathbb{I}(h(x_i)=k, y_i=k)\right)
\tag{11}
$$

$$
= \sum_{k=1}^{K} P(y=k)acc(y=k) = \mathbb{E}_{Y \sim P(Y)}\left[acc(Y)\right]
$$

As shown in eq. 11, the accuracy metric is equal to the expectation of accuracy w.r.t to the distribution of labels, $P(Y)$. When $P_s(Y) \neq P_t(Y) = \mathbb{U}(1/K)$, training with imbalanced data maximizes accuracy w.r.t $P_s(Y)$ where large classes dominate, while on test dataset the accuracy metric calculates the expectation of accuracy w.r.t a uniform distribution . Consequently, training with imbalanced

---

**Algorithm 2:** Binary Search Algorithm for $\lambda$

---

**Data:** $\{x, y\} \in D_{val}$

**Result:** $\lambda$

  **Initialize:** $L \leftarrow 0.0$, $H \leftarrow 2.0$, $prec \leftarrow 0.1$ *metric()*      ▷ *metric()* takes a $\lambda$ and evaluates a model on $D_{val}$

  **if** *metric(0.1) < metric(0.0)* **then**

      **Return** 0.0                                 ▷ $\lambda = 0.0$ is the optimal solution

  **while** *metric(H)* $\geq$ *metric(0.0)* **do**

      $H+ = 5 * prec$            ▷ Enlarge the search range s.t. the optimal $\lambda$ is absolutely contained

  **procedure** FINDMAXLAMBDA(*metric()*,$L$, $H$)

    **if** $L == H$ **then**

      **Return** $L$                                   ▷ Reached the last $\lambda$

    **if** $H == L + 1$ **then**

      **if** *metric(L)* $\geq$ *metric(H)* **then**

        **Return** $L$

      **else**

        **Return** $H$                          ▷ Reached the last two $\lambda$s

    $M \leftarrow (L + H)/2$

    **if** *metric(M) > metric(M + prec) and metric(M) > metric(M − prec)* **then**

      **Return** $M$

    **if** *metric(M) > metric(M + prec) and metric(M) < metric(M − prec)* **then**

      **Return** FindMaxLambda(*metric()*,$L$,$M − prec$)     ▷ Optimal $\lambda$ lies on the left side

    **else**

      **Return** FindMaxLambda(*metric()*,$M + prec$,$H$)     ▷ Optimal $\lambda$ lies on the right side

  **Complexity:** $\mathcal{O}(\log N)$ where $N = (H − L)/prec$

---

data often biases towards large classes to maximize eq. 11 and often results in poor performance on uniform evaluation data.

**Implicit label prior shift** is unconventional because it happens when $P_s(Y) = P_t(Y) \neq \mathbb{U}(1/K)$ . This is the case for semantic segmentation, where classes such as pedestrians, poles, etc. occupy a much smaller portion of an image in both training and test data. However, people tend to assign equal importance to all classes regardless of their actual pixel percentage. This is reflected by the popularity of class average metrics such as mean accuracy and mean Intersection over Union (mIOU). In other words, label prior shift is implicitly caused by using class average metrics even though the test distribution is equal to training. Let's look at the another popular metric, mean accuracy, which is often used when the test data is not uniform (in contrast to the first case).

$$mACC(X, Y) = \frac{1}{K} \sum_{k=1}^{K} \frac{1}{N_k} \sum_{i=1}^{N} \mathbb{I}(h(x_i) = k, y_i = k) \tag{12}$$

$$= \sum_{k=1}^{K} \frac{1}{K} \left( \frac{1}{N_k} \sum_{i=1}^{N} \mathbb{I}(h(x_i) = k, y_i = k) \right) = \sum_{k=1}^{K} \frac{1}{K} acc(Y = k) = \mathbb{E}_{Y \sim \mathbb{U}(1/K)} \left[ acc(Y) \right]$$

As shown in eq. 12, mean accuracy is equal to the expectation of accuracy w.r.t a uniform label distribution. Mean accuracy is exactly the same as the accuracy metric on uniform test data in the first case. In other words, evaluating on non-uniform test data with mean accuracy is equivalent to evaluating on uniform test data with accuracy in expectation. Even though the training distribution is the same as the test distribution and both are non-uniform, using mean accuracy results in an implicit label prior shift.

## 6.4 Hyperparameter for Cifar Experiments

Table 6 documents the hyperparameters used for the Cifar experiments in table 2 in the main paper. The search for $\lambda$ is very efficient because it dose not require training. The modified binary search algorithm in Appendix Sec. 6.2 can be used to find a $\lambda$ in $\mathcal{O}(\log N)$ time.

Table 6: **Top1 validation error↓ for Imbalance Calibration**. CE is a baseline trained with unmodified cross-entropy loss. CB refers to class balanced loss in [1]. The subscript denotes the $\lambda$ values for the experiments

| Dataset | Imbalanced CIFAR-10 | | | | Imbalanced CIFAR-100 | | | | |
|---|---|---|---|---|---|---|---|---|---|
| Imbalance Type | long-tailed | | step | | long-tailed | | step | | AVE |
| Imbalance Ratio | 100 | 10 | 100 | 10 | 100 | 10 | 100 | 10 | |
| CE-IC (Ours) | $20.14_{2.1}$ | $11.37_{0.7}$ | $25.61_{1.1}$ | $11.34_{1.0}$ | $59.08_{1.3}$ | $41.91_{1.0}$ | $53.48_{1.2}$ | $40.64_{1.3}$ | 32.95 |
| Focal-IC (Ours) | $19.84_{1.3}$ | $11.99_{1.1}$ | $25.68_{0.8}$ | $\mathbf{10.99_{1.0}}$ | $58.35_{1.1}$ | $42.01_{1.0}$ | $\mathbf{52.73_{1.2}}$ | $40.70_{1.0}$ | 32.79 |
| CE-DRW-IC (Ours) | $\mathbf{18.91_{1.5}}$ | $11.44_{1.2}$ | $\mathbf{21.45_{1.2}}$ | $11.42_{1.0}$ | $\mathbf{56.89_{0.6}}$ | $\mathbf{41.44_{0.5}}$ | $54.40_{0.3}$ | $\mathbf{40.07_{0.3}}$ | $\mathbf{32.00}$ |
| Focal-DRW-IC (Ours) | $21.63_{1.0}$ | $11.80_{0.5}$ | $21.63_{1.0}$ | $11.48_{0.5}$ | $59.43_{0.5}$ | $43.35_{0.0}$ | $56.89_{0.6}$ | $41.93_{0.0}$ | 33.52 |

Figure 3: **Hyperparameter $\lambda$ search on iNaturalist2018 (Left) and Synthia (Right) validation set for imbalance calibration**. $\lambda = 0.0$ corresponds to the baseline performance. Note that in the main paper, we report 3-fold cross validated resutls for iNaturalist2018. Here we show the search on the entire validation set. $\lambda = 0.4$ is used as the parameter for subsequent experiments on Synthia fusion experiments in sec. 4.4.2

## 6.5   Lambda Search on Validation Set for iNatrualist2018 and Synthia

The left side of Fig. 3 shows the performance curve of top1 and top5 accuracy on iNatrualist2018 with different $\lambda$. The right side of Fig. 3 shows the performance curve of both mean IOU and mean accuracy on a Synthia validation set with varying $\lambda$ values. On the far right where $\lambda = 1.0$ we obtain the performance of the rebalanced posterior as in eq. 5 and on the left we recover the performance of the discriminative model. Intuitively, as we move towards the full rebalanced posterior by increasing $\lambda$, the algorithm gives more weight to minority classes and the mean accuracy continues to increase. However this comes at the cost of significant increase in false positives as the mean IOU drops. The trade-off between mean IOU and mean accuracy reflects the imperfect learning of the dataset discussed in sec. 3.2. By searching for a good balance, it is possible to improve mean accuracy while maintaining a good mean IOU. The performance curves in both experiments exhibit concavity as in the simpler 2D toy example 4.1. This further validates our proposed binary search algorithm in Appendix Sec. 6.2 which relies on the assumption of concavity.

## 6.6   Qualitative Results for Synthia RGBD fusion Experiments

Fig. 4 shows qualitative results for the Synthia fusion experiments in Sec. 4.4.2. Our method UNO-IC is able to capture small details such as pedestrians in the distance even under severe visual degradations. Because the rain condition is not encountered during training, the rgb channel failed to cope with the degradation. UNO [19] dynamically shifts the fusion algorithms weight to the depth channel while our imbalance calibration method gives more weights to small objects. The qualitative and quatitative results in Sec 4.4.2 demonstrate the effectiveness of our unifed perspective 3.1 and Alg. 1.

Figure 4: **Qualitative results for Synthia Rain**. The rain condition is not seen during training. Our algorithm UNO-IC can capture fine details in segmentation even under severe viusal degradation not encounted during training. Our method combined with UNO demonstrates is effective against extreme dataset imbalance and unseen degradations in semantic segmentation.