[Reviews · NeurIPS 2020]

Review 1

Summary and Contributions: This paper introduces a post-processing algorithm for re-calibrating a classifier in the presence of label imbalance. The algorithm is very efficient and relies only on tuning a single hyperparameter that interpolates between the unbalanced predictor and the rebalanced posterior predictor. Through extensive experiment on common benchmark datasets, the authors show that their method can consistently reach or outperform SOTA, especially when combined with other rebalancing techniques.

Strengths: 1. In contrast to other rebalancing techniques that modify the loss or training procedure, the proposed method operates as a post-processing step after training, and only requires tuning a single hyperparameter on a validation set. This is very beneficial from an adoption standpoint. 2. The method is principled and is inspired by rebalancing the Bayes optimal classifier. As far as I know, it is also novel compared to prior work on rebalancing. 3. The empirical evaluation is convincing. The authors evaluated on standard benchmark datasets and showed that this simple modification can effectively correct the label imbalance problem when applied to standard learning with cross entropy loss. The post-processing nature of this method also allows it to be combined with other rebalancing techniques to achieve further improvements.

Weaknesses: There seem to be some discrepancies in the reported result from prior work. In Table 1, LDAM-DRW on CIFAR-100 with step imbalance ratio 10 is reported to be 43.49, while in the original paper it is 40.54. Similarly, in Table 2, the authors report LDAM-DRW to achieve an accuracy of 35.42, compared to 32.00 in the original paper. Although this change would mean that CE-IC is only competitive with LDAM-DRW rather than outperforming in those settings, the benefit of post-processing still renders CE-IC as a superior method.

Correctness: Empirical methodology followed standard settings of prior work on rebalancing and is sound.

Clarity: I find that the paper is organized and written clearly, and I had no trouble understanding the technical details.

Relation to Prior Work: The method contrasts itself against prior work that require modified loss or training procedure, and I find that their similarities and differences are adequately discussed.

Reproducibility: Yes

Additional Feedback: The rebalancing algorithm carries significant resemblance to confidence calibration using temperature scaling; see Chuan Guo, Geoff Pleiss, Yu Sun, and Kilian Q Weinberger. On calibration of modern neural networks. In Proceedings of the 34th International Conference on Machine Learning-Volume 70, pages 1321–1330. JMLR. org, 2017. I am wondering if there are implications to improving the calibration error of the trained classifier? A strong answer to this question could further strengthen the proposed method as a confidence calibration method for label-imbalanced data. ------------------------------------------------------ Another reviewer suggested the reference: Saerens et al. Adjusting the Outputs of a Classifier to New a Priori Probabilities: A Simple Procedure. Neural Computation, 2002. It appears to me that the posterior rebalancing algorithm proposed in this paper is conceptually similar to the above prior work (under the known a priori setting), which diminishes the novelty of this work. I am reducing my rating to 7 as a response.


Review 2

Summary and Contributions: This paper proposes a method to obtain better performance under label prior shift (class balance differed between train/test) and non-semantic shift (class-conditional probability differs between train/test) scenarios. The proposed method is based on minimizing the linear combination between two KL-divergences of the source posterior probability and the target posterior probability. Experiments on several datasets and network architectures are also provided.

Strengths: The paper has the good motivation and the proposed method is relatively easy to implement. The authors kindly provided the Intuition to illustrate how the proposed method can be effectively useful, e.g., in practice, using the direct application of Thm.1 may suffer from too large class balance ratio and it may be more robust to use a posterior probability obtained from minimizing the linear combination of source/test posterior probabilities.

Weaknesses: 1. This paper discussed related work that are mostly about learning under class imbalance. But I think the main focus of this paper is learning under class prior shift, not under class imbalance. Thus, related work should contain more information about learning under label prior shift or domain adaptation. Or perhaps it is important to note that class imbalance is not the only cause for label prior shift. 2. I am convinced that the proposed method should be better than straightforwardly using the discriminative source posterior since class prior shift occurs. But I don't see how the proposed method addresses the problem of non-semantic shift theoretically, or it is only showed empirically? Plus, it seems the assumptions of label prior shift and non-semantic likelihood shift contradict each other, i.e., label prior shift assume the class-conditional probabilities are identical but non-semantic likelihood shift focuses on when the class-conditional probabilities are different. 3. Since this paper focuses on class prior shift, the comparisons under class prior shift scenario with class prior shift method are somewhat lacking. On the other hand, methods for learning under class imbalance (but do not consider class prior shift) are used instead. 4. Clarity (please see the clarity section).

Correctness: I think the proposed method is sound.

Clarity: I found some parts are a bit difficult to follow and I have the following questions: 1. How to get P_t(y) in this paper? I think it is clearer to clarify the problem setting in Sec. 3.1, e.g., what is the input of the problem (labeled training data and labeled test data?) Also how to get P_s(y)? I guess it is from counting the training data with its associated labels. It is kind to the readers to write it down (or I may have missed it...). 2. I couldn't find it but what are the train/test proportions for each dataset in the experiments? 3. Theorem 1 is very important in this paper and I think how to derive it may add more understanding to the readers (it seems the proof was not that long to add it in the main body). 4. In Sect. 3.3, I found it is quite difficult to understand why doing so helps mitigating the problem of non-semantic shift. 5. At the beginning of Theorem 1, is the first sentence: "given that h_s(x) is the Bayes classifier on P_s(X,Y)": is this sentence has any relationship with this theorem, I don't see how it's used in Theorem 1 anywhere. 6. In Algorithm 1, to make a paper self-contained, it is necessary to add more information about UNO, in particular, what is the form of \delta_m(x). 7. The full search algorithm for lambda is also in the appendix. Since it seems quite simple it may be possible to write it clearly in the main body and I think it is quite useful to also emphasize that to calculate the score of lambda for each trial is sufficient because it does not require training, as stated in the appendix. Minor comments: 1. line 170 ( detailed -> (detailed 2. I think it is better and important to use the abbreviation consistently for the professional writing. For example, "Fig. 1" at line 180 and "fig. 1" at line 182 should be unified. More examples are (Eq. vs eq.), (sec. and Sec.) and (Theorem vs theorem). They are used interchangeably in a quite random manner in this paper. 3. line 174: space between Eq. and 8 Eq. (9): how about using \times or \cdot instead of * ? 4. line 289: lambda -> $\lambda$ 5. line 292: underylying -> underlying 6. Eq. 1: is it a typo to use f_y instead of f_t? 7. line 111: f_t(x|y) is the "test" class conditional proba.... P_t(y) is the "test" prior distribution. I think it is kind to reads to emphasize that it is with respect to the test distribution.. 8. appendix line 378: abbreivate -> abbreviate 9. Figure 2: marjority -> majority. S $\lambda$ increases <-- was "S" a typo? 10. appendix Figure 3: resutls -> results 11. appendix 7.7: Cifar -> CIFAR 12. appendix 7.8: iNatrualist2018 -> iNaturalist2018 13. appendix 7.4: binary searcy -> binary search 14. appendix line 466: k-comnination 15. appendix line 487: dose -> does (there are more typos and small writing issues in the appendix so please fix it) 15. Table 3: subscript value for lambda seems to be missing. Overall I feel a strong necessity of reading the supplementary materials and some existing work to understand this paper. I think at least the proposed method and problem setting should be clearly stated in the main body. On the other hand, some experiments that might not give more insightful message given already presented experiments may be moved to appendix because they are quite space-consuming.

Relation to Prior Work: More discussion on class prior shift and domain adaptation are needed if the main goal is to handle label prior shift.

Reproducibility: Yes

Additional Feedback: I believe the assumption of class prior shift that this paper used contradicts with the non-semantic likelihood shift assumption, i.e., class-conditional should remain unchanged for the class prior shift, but they need to be different in likelihood-shift. Thus, dealing with both shift at the same time is somewhat close to a general assumption domain adaptation, i.e., test and train distributions are different. Thus, the assumption for deriving the rebalanced posterior would be invalid. If I understood correctly, it is important to emphasize on this point. Question: 1. Does Theorem 1 has any relationship with the adjusted posterior proposed in Saerens et al., 2002 [1]? 2. How many methods use the test class prior information in the experiment? 3. How do you split the validation set for CIFAR dataset? And do you use validation set for other compared methods or use training+validation for other methods? 4. Were all experiments only focus on the class imbalance scenario and no label prior shift with varying number of shifting are considered? Finally, I would like to highlight that the lack of comparison with the methods for learning under class prior shift and the clarity of this paper are the main concerns that influence my overall score the most. [1] Saerens et al. Adjusting the Outputs of a Classifier to New a Priori Probabilities: A Simple Procedure. Neural Computation, 2002. Finally, there might be a chance I failed to acknowledge some insightful parts of this paper. Thus, I will read the rebuttal carefully and if my concerns are addressed or they are irrelevant because of my misunderstanding, I am happy to increase the score. ===== After the author's response I have read the reviews and the author's response. I appreciate the authors for addressing my concerns. I have decided to slightly increase my score from 4 to 5. And I would like to suggest the authors to improve the clarity of the paper to make the paper more self-contained if this paper is accepted. 1. As [2] also suggests that this problem is "a special case" of domain adaptation. But I agree and understand that label prior shift is just a small instance of domain adaptation and label prior shift can be said to be easier because we assume how the distribution shifts. [2] also clearly stated that they assume that P_t is known and it is balance. Nevertheless, this is not a big issue. 2. I also asked about how to obtain P_t and what kind of P_t was used but my question was not addressed. But I am also aware of the fact that the author's response has limited space. I guess P_t is balanced. 3. About the contradiction of assumption. I am sorry if my question was not clear. I understood that both assumption are not contradicting each other if it is phrased as the authors suggested. But usually label shift assumes that the class conditional probabilities of train/test are identical. My point that Theorem 1 hugely relies on the assumption that the test distribution has the same class conditional likelihood as the training set, i.e., non-semantic shift does not hold. Thus, theorem 1 will not hold when considering non-semantic shift. This point should be made clear in the paper. 4. About the relationship with Saerens et al. 2002: Thank you. The explanation is clear. 5. About "given that h_s(x) is the Bayes classifier on P_s(X,Y)" at the beginning of Thm.1: Thanks for the answer. But still, I think this sentence can be removed from the theorem because it seems h_s(X) is not used in the theorem as far as I understood (although it is used in other parts in the paper).


Review 3

Summary and Contributions: The paper focuses on classification, in a long-tailed learning setup. I.e. where training class distribution is imbalanced. The paper main contribution is in providing a post-processing scheme that calibrates a classifier softmax scores, basically by suggesting p_{calibrated}(class|image) \prop p_{pretrained}(class|image)*(p_{train}(class))^{-\lambda}, where \lambda is a hyper-param selected with cross validation, and p_{pretrained}(class|image) are the softmax scores of a pretrained network. Tuning \lambda allows to trade-off precision-recall like metrics. Overall, the paper suggest a simple and effective post-processing step that is equivalent or better then current SoTA approaches for long-tail learning, that does not requires specialized training. I have some criticism about the hypothesis that lead to the proposed calibration scheme, the clarity of some parts of the paper, and about some of the benchmarks. I recommend borderline-reject, however I am open to positively update my score after the rebuttal.

Strengths: + Simple + Allows to reuse a network pretrained with a standard cross-entropy loss + Empirical results are equivalent to current SoTA + Allows to apply with semantic-segmentation tasks.

Weaknesses: -- The motivation that lead to the proposed calibration technique is lacking (see my comments under "Correctness") -- I found it hard to understand section 3.3 (see "Correctness/Clarity") -- Lacking comparison to all SoTA methods (see "Prior work") -- I am unsure about the comparison over SYNTHIA benchmark (have the authors compared to all baselines?)

Correctness: -- It is not clear *why* to balance P_r(y|x), which reweighed for the target class distribution, with P_d(y|x) which is fitted on the source class distribution. Luckily the proposed balancing expression leads to a simple expression that can tune the class priors using a hyper-param (lambda) such that it trades-off "precision-recall". However, the motivation is lacking, since we would ideally like to make predictions according to the true target distribution. -- It is hard to understand section 3.3. I couldn't verify its correctness

Clarity: Some parts are clear, other should be improved (see details under my comments about correctness).

Relation to Prior Work: -- The paper should compare iNaturalist with the [Decoupling] method which is the current SoTA. -- I am not familiar with the SYNTHIA benchmark. However it was introduced in 2016. Aren't there other baselines to compare with except UNO? [Decoupling] Kang et al, Decoupling Representation and Classifier for Long-Tailed Recognition, ICLR 2020 (accepted & online since December 2019)

Reproducibility: Yes

Additional Feedback: Post rebuttal comments: I thank the authors for their answers. Following the rebuttal I am more convinced about the experimental results. But I am still unsatisfied about the authors answer about my question "*why* to balance P_r(y|x)?". The authors replied that "we do not have access to the true target distribution, ". However, in practice, all the balanced accuracy metrics are equivalent for having a uniform target distribution. And respectively, the method exploits this fact when it uses Pt as a balanced (uniform) target distribution. Therefore, I have decided to keep my score as is (5)

[Author Response · NeurIPS 2020]

| | CE | Focal[1] | CB[2] | LDAM[3] | BNN[4] | LWS[5] | CE-IC (ours) | CE-DRW-IC (ours) |
|---|---|---|---|---|---|---|---|---|
| Resnet50 | 38.00 | 38.33 | 38.88 | 35.42 | 33.71 | 34.10 | **32.16 ± 0.41** | **32.06 ± 0.38** |

Table 1: **Validation error↓ on iNaturalist2018.** [1] Focal,Tsung-Yi Lin et al (ICCV 2017); [2] CB, Cui Yin et al (ICCV'2019); [3] LDAM, Kaidi Cao et al (NIPS'2019); [4] BBN, Boyan Zhou et al (CVPR'2020); [5] Decoupling, Kang et al (ICLR' 2020). All methods use resnet 50 and are trained for 90 epochs.

We thank the reviewers for their thorough reviews and positive comments about the novelty, effectiveness and adaptability
of the method. We will make corresponding changes to reflect the comments.

**Paper Summary**    **R2: Our main contribution (Sec.3.2) is a rebalance method for class imbalance which is a**
**specific (and difficult) form of label prior shift, and is not domain adaptation; label prior shift in the paper**
**refers specifically to change in empirical class frequencies between source/target distribution.** Domain adaption
usually dose not consider class imbalance explicitly. We follow the standard datasets and works within this subfield
[2][3][4][5], exhibiting *only* label prior shift (line 113-117). Specifically, $P_s(Y)$ is the source class priors obtained
by counting the number of examples from each class in the training data and $P_t(Y)$ is the class priors for the testing
data. The definition of *label prior shift* (line 112-113) and *non-semantic likelihood shift (NSLS)* (line 119-121) do not
contradict each other because they constitute different parts of a joint distribution and can occur simultaneously. When
both shifts occur, optimality is not guaranteed (which we will mention in the paper) but empirically the performance is
still strong (Exp. Sec.4.4.2). **Our second contribution (Sec.3.3) is demonstrating the *adaptability* of the rebalance**
**method by combing it with a *multi-modal fusion algorithm*.** Unlike domain adaptation, we don't assume to have
unlabeled data in the target domain during training. The fusion algorithm deals with NSLS by *weighting* the modality
affected by NSLS whereas domain adaption aims to *adapt* to domain shift utilizing additional target domain data.

**Does it also calibrate the network?**    **R1:** The imbalance calibration technique (Sec.3.2) *rebalances* a network.
However, the probabilistic nature means that it can be combined with other probabilistic techniques. One reason of
combining with UNO (Sec.3.3) is to demonstrate the **adaptability** of this method since UNO uses the exactly same
temperature scaling technique for calibrating a network.

**More Comparisons to SOTA**    **R2 R3:** Our paper proposes a rebalance method (sec.3.2) for class imbalance and
compared to those SOTA methods. We thank the reviewers for pointing out works we missed and report them here. We
believe that **our method is more effective and easier to adapt than current SOTA methods** as shown in table 1.

**Inclusion of non-semantic likelihood shift (NSLS)**    **R2 R3:** To clarify, Section 3.3 is meant to demonstrate that our
rebalance technique (Sec.3.2) is general, and can be *combined* with existing probabilistic methods for mutli-modal
fusion e.g, UNO *by considering class imbalance in semantic segmentation*. The mechanism by which the fusion
algorithm deals with NSLS is temperature scaling. In a mutli-modal fusion setting, when one modality is under NSLS
its prediction is no longer reliable. The fusion algorithm flattens the distribution affected by NSLS and effectively
diminishes its contribution when fused with other distributions. It resembles a conventional "gating" mechanism which
filters out the degraded modality. We realize that this section requires more background knowledge will add proper
introduction and expand explanation in the main paper.

**Clarification on Theorem 1**    **R2:** The first sentence "Given that $h_s(x)$ is the Bayes classifier ..." is equivalent to
"Given $P_s(Y|X)$ is the posterior distribution of the source dataset" because in Eq.2, $h_s(x)$ is defined in terms of
$P_s(Y|X)$. While Saerens et al.2002 arrives at the same well-known equation through Bayes Rule as we did, Theorem
1 proved its **optimality** through Bayes Risk. The major contribution of Saerens is an EM algorithm to estimate the
unkonwn $P_t(Y)$. **R3:** We agree that if we have the true target distribution, the problem can be solved. However
knowing that *we do not have access to the true target distribution*, we propose to find a better approximation to
the true distribution. **Intuitively speaking, $P_r(y|x)$ often *over-emphasizes small classes* while $P_d(y|x)$, which fits**
**the imbalanced source data, naturally *biases towards large classes* on test data (line 150-152).** This observation
motivates us to balance the two posteriors. Therefore, Hypothesis 1 states that **a better approximation to the true**
**target distribution** can be found by trading off $P_r(y|x)$ and $P_d(y|x)$ through optimization. It is validated empirically
through subsequent experiments since varying $\lambda$ yields better performance.

**Clarification on Experimental Setting**    **R1:** we report our re-trained model performance for other methods using
exactly the same code from the authors. The performance is consistent with other papers using the same code. Even
considering the original performance, our method still outperforms other methods. **R2:** For iNatrualist and CIFAR
datasets, we use the official train/validation splits. For Synthia datasets we split the train/test/validation according to the
$7 : 2 : 1$ ratio. For CIFAR experiments we report the validation errors as in other papers and use the the same train and
validation for all methods. **R2 R3:** Synthia is a semantic segmentation dataset and uses a very different architecture
and training schedule compared to image classification datasets. **For fair comparison, we compared to imbalance**
**losses but not methods requiring significant changes to training strategy or architecture (line 268-269) since**
**most imbalance methods are developed for image classification but not semantic segmentation.** This shows that
our rebalance method is more ***general*** and applicable to a broader range of problems than current methods.

[Meta-Review · NeurIPS 2020]

This paper addresses the prior shift between training and testing scenarios. Using the optimal Bayes classifier, they derive a factor which is optimizes on a validation set. Reviewers found the method novel, simple to understand, easy to implement and applicable to a wide variety of tasks. It can also be combined easily with other existing domain adaptation methods. On several benchmarks, authors have achieved improved on state-of-the-art results. Thus demonstrated the general applicability of this approach. Reviewers #2 and #3 provide constructive suggestions to improve the presentation and the value of this work.